# Rapid Whole-Body FDG PET/MRI in Oncology Patients: Utility of Combining Bayesian Penalised Likelihood PET Reconstruction and Abbreviated MRI

**DOI:** 10.3390/diagnostics13111871

**Published:** 2023-05-26

**Authors:** Junko Inoue Inukai, Munenobu Nogami, Miho Tachibana, Feibi Zeng, Tatsuya Nishitani, Kazuhiro Kubo, Takamichi Murakami

**Affiliations:** 1Department of Radiology, Kobe University Graduate School of Medicine, 7-5-1, Kusunoki-cho, Chuo-ku, Kobe 650-0017, Hyogo, Japan; 2Department of Radiology, Kobe University Hospital, 7-5-2, Kusunoki-cho, Chuo-ku, Kobe 650-0017, Hyogo, Japan; 3Division of Medical Imaging, Biomedical Imaging Research Center, University of Fukui, 23-3, Matsuokashimoaizuki, Eiheiji, Yoshida 910-1193, Fukui, Japan

**Keywords:** PET/MRI, image reconstruction, fluorodeoxyglucose, Bayesian penalised likelihood, whole-body imaging, abbreviated MRI

## Abstract

This study evaluated the diagnostic value of a rapid whole-body fluorodeoxyglucose (FDG) positron emission tomography (PET)/magnetic resonance imaging (MRI) approach, combining Bayesian penalised likelihood (BPL) PET with an optimised β value and abbreviated MRI (abb-MRI). The study compares the diagnostic performance of this approach with the standard PET/MRI that utilises ordered subsets expectation maximisation (OSEM) PET and standard MRI (std-MRI). The optimal β value was determined by evaluating the noise-equivalent count (NEC) phantom, background variability, contrast recovery, recovery coefficient, and visual scores (VS) for OSEM and BPL with β100–1000 at 2.5-, 1.5-, and 1.0-min scans, respectively. Clinical evaluations were conducted for *NEC_patient_*, *NEC_density_*, liver signal-to-noise ratio (SNR), lesion maximum standardised uptake value, lesion signal-to-background ratio, lesion SNR, and VS in 49 patients. The diagnostic performance of BPL/abb-MRI was retrospectively assessed for lesion detection and differentiation in 156 patients using VS. The optimal β values were β600 for a 1.5-min scan and β700 for a 1.0-min scan. BPL/abb-MRI at these β values was equivalent to OSEM/std-MRI for a 2.5-min scan. By combining BPL with optimal β and abb-MRI, rapid whole-body PET/MRI could be achieved in ≤1.5 min per bed position, while maintaining comparable diagnostic performance to standard PET/MRI.

## 1. Introduction

The advantage of using positron emission tomography (PET)/magnetic resonance imaging (MRI) for the assessment of oncology patients is its capability for simultaneous acquisition and high-contrast resolution MRI [1,2]. However, a major challenge of PET/MRI in oncology is the lengthy examination time, which is due to the whole-body PET scan and regional MRI with multiple sequences. This extended examination time can reduce the throughput of clinical examinations and cause discomfort for patients during scanning. To improve the clinical utility of PET/MRI in oncology patients, it is essential to accelerate the examination process.

The use of Bayesian penalised likelihood (BPL) reconstruction, also known as Q.Clear, allows for full convergence without image degradation, which cannot be achieved by standard ordered subset expectation maximisation (OSEM) reconstruction for the exact duration of the emission scan [3,4,5,6]. Therefore, BPL is useful in improving the image quality of low-count PET images acquired through low-dose administration and/or short emission times.

Regarding MRI, the concept of an abbreviated MRI (abb-MRI) is being extensively studied. Abb-MRI is a shortened version of standard MRI (std-MRI) that uses fewer sequences. Recent works have shown that abb-MRI, which acquires only the minimum necessary sequences, equals the diagnostic performance of std-MRI across studies of breast cancer, liver tumours, prostate cancer, and other conditions [7,8,9,10]. Therefore, in PET/MRI, abb-MRI implementations can bring rapidity into whole-body MRI.

We hypothesised that combining BPL with an optimal beta (β) value and abb-MRI would enable rapid whole-body PET/MRI with a diagnostic performance equivalent to conventional PET/MRI with OSEM in standard emission time and standard whole-body MRI protocol. Thus, the study had two main purposes: (a) to assess the optimal β value of time-of-flight (TOF) BPL in a short (1.0 min and 1.5 min) emission scan duration (BPL_1.0_ and BPL_1.5_, respectively), equivalent to TOF-OSEM reconstruction in a standard (2.5 min) emission scan duration (OSEM_2.5_) in both phantom and clinical evaluations, and (b) to evaluate the diagnostic performance of the combination of BPL with optimal β values and abb-MRI in lesion detection and differentiation between malignant and benign, as compared to that of OSEM_2.5_ and std-MRI.

## 2. Materials and Methods

### 2.1. PET/MRI

The hybrid PET/MRI scanner used in this study was a SIGNA PET/MR by GE Healthcare, operating at a magnetic field strength of 3.0 T. The whole-body imaging was performed using a 19-channel head and neck coil, a 16-channel anterior array coil, and a 16-channel central molecular imaging array coil. The PET component of the scanner utilised silicon photomultipliers and detectors capable of TOF PET with a timing resolution of fewer than 400 ps [11]. PET images were reconstructed using TOF-BPL with various β values and TOF-OSEM with a Gaussian filter of 4.0 mm, two iterations, and 16 subsets.

### 2.2. Phantom Study

A National Electrical Manufacturers Association (NEMA) image quality body phantom was used to evaluate the difference in image quality between short and standard emission times. The background of the phantom was filled with approximately 5.30 kBq/mL of ^18^F-fluorodeoxyglucose (FDG), and the spheres were filled with a radioactivity concentration four times higher. The phantom preparation and subsequent data evaluation followed the Japanese guidelines for the oncology FDG PET/computed tomography (CT) data acquisition protocol [12].

The emission scan lasted for 30 min in list mode, and the data were sorted to obtain three images for the scan duration, starting at 1.0, 1.5, and 2.5 min. The PET list mode reconstructions were performed for ten different β values (100, 200, 300, 400, 500, 600, 700, 800, 900, and 1000).

The noise-equivalent counts for the phantom (*NEC_phantom_*) were measured at 2.5-, 1.5-, and 1.0-min emission scan durations. The background variability (*BV*), contrast recovery (*CR*), and recovery coefficient (*RC*) of the 10-mm sphere were measured by reconstructing images with OSEM and BPL with β100–1000. The *NEC_phantom_* was calculated using the following equations:(1)$$NEC_{phantom}={\left(1-SF\right)}^{2}\frac{{\left(T+S\right)}^{2}}{\left(T+S\right)+\left(1+k\right)fR}\;\;\mathrm{[Mcounts]}$$
(2)$$f=\frac{S_{a}}{\pi r^{2}},$$
where *T*, *S*, and *R* represent the true, scattered, and random coincidences, respectively, acquired during the emission time. *SF*, *k*, and *f* represent the scatter fraction, random scaling factor, and the ratio of the object size to the axial scanning field of view, respectively. *S_a_* and *r* represent the cross-sectional area of the phantom and the radius of the detector ring diameter, respectively. SF is an intrinsic value based on the NEMA NU-2 standard [11].

The *BV* was calculated using the equation
(3)$$BV=\frac{SD_{10\mathrm{mm}}}{C_{B,\;10\mathrm{mm}}}\times 100\;\left[\%\right],$$
where *C_B_*_;10mm_ is the mean activity for the 10-mm ROIs in the background area and *SD*_10mm_ is the standard deviation of the mean activity for the background 60 ROIs.

The *CR* was calculated using the equation
(4)$$CR=\frac{Q_{H,\;10\mathrm{mm}}}{BV},$$
(5)$$Q_{H,\;10mm}=\frac{\frac{C_{H,\;10\mathrm{mm}}}{C_{B,\;10\mathrm{mm}}}-1}{\frac{\alpha_{H}}{\alpha_{B}}-1}\times 100\;\left[\%\right].$$
where *C_H_*_;10mm_ and *C_B_*_;10mm_ are the mean activity in the ROI for the 10-mm sphere and the mean activity in all the background 10-mm ROIs, respectively, and *α_H_*/*α_B_* is the activity concentration ratio of the hot sphere to the background.

*RC* was calculated using the equation
(6)$$RC=\frac{C_{10\mathrm{mm}}}{C_{37\mathrm{mm}}},$$
where *C*_10mm_ and *C*_37mm_ are the maximum activities of the 10-mm and 37-mm diameter hot sphere, respectively.

Additionally, three readers assessed the image quality using a 5-point visual score from 0 to 4. The sum of the 3-point scores for the delineation of the 10 mm diameter hot sphere (ranging from 0 to 2) and that for the background homogeneity (ranging from 0 to 2) were used to calculate the visual scores. The visual scores were determined by comparing them with OSEM_2.5_.

The range of optimal β values in the phantom experiment was determined as follows: for BV, CR, and RC, the candidate β values were those that were better than OSEM_2.5_ or the top three β values that were close to OSEM_2.5_; for the visual score, the candidate β values were those where at least one of the three readers rated better than OSEM_2.5_. Finally, four consecutive β values were obtained from the set of β values that satisfy at least two of the BV, CR, and RC criteria, or from the set of β values that were candidates for the visual score.

### 2.3. Clinical Evaluation

#### 2.3.1. Patients

The retrospective study was approved by the Institutional Review Board, which waived the requirement for informed patient consent. All patients fasted for at least six hours before the examination. The inclusion criteria for the assessed patients in the clinical evaluation were patients with current or previous malignancy who had undergone a PET/MRI scan and whose final diagnosis was confirmed by histopathological results and/or follow-up PET/MRI and PET/CT scans during the 6-month follow-up period. The following exclusion criteria were applied for clinical evaluation: patients under the age of 20. Whole-body PET/MRI examinations were performed 60 min after intravenous injection of 3.5 MBq/kg ^18^F-FDG.

#### 2.3.2. Whole-Body PET/MRI Protocol

The whole-body PET/MRI protocol comprised five to six bed positions, including one to two beds having respiratory-gated PET/MRI in the thoracic and upper abdominal regions (Figure 1). The long axial field of view (FOV) length was 25 cm, with 89 slices per bed position, which included an overlap of 24 slices (27%). The gated PET scans required twice the scan duration of non-gated beds and were reconstructed using 50% of the data. The gated MRI acquisition was performed using the respiratory bellows and/or navigator-echo method. No intravenous contrast-enhancing material was administered for the MRI.

Attenuation correction for PET was conducted through the generation of an MRI-based μ-map by the vendor, which was acquired simultaneously for 10 s with a two-point Dixon three-dimensional volumetric interpolated fast spoiled gradient echo (Dixon) sequence under free-breathing (MRAC).

Each PET/MRI bed position included two MRI sequences: a standard protocol (std-MRI) and an abbreviated protocol (abb-MRI). The std-MRI consisted of T1-weighted Dixon and three-dimensional fast spin-echo T2-weighted images (3D-T2WI) for 2.5-min PET emission scans. The abb-MRI protocol only used T1-weighted Dixon for 1.5-min and 1.0-min PET emission scans. Dixon is a three-dimensional dual-echo gradient echo (GRE) sequence that uses a two-point Dixon method for water-fat separation. The parameters for Dixon were as follows: repetition time (TR) of 3.9 ms, first echo time (TE) of 1.1 ms, second TE of 2.2 ms, slice thickness of 4.0 mm, flip angle (FA) of 12°, number of excitations (NEX) of 1, matrix size of 200 × 288, FOV of 45.0 cm with 80% phase field of view, and an estimated scan time of 35 s. The parameters for 3D-T2WI were TR of 1800 ms, TE of 90.0 ms, echo train length of 96, slice thickness of 4.0 mm, FA of 90°, NEX of 1, matrix size of 256 × 224, FOV of 35.0 cm, and an estimated scan time of 62 s.

#### 2.3.3. Clinical Evaluation for Optimal β Value

To determine the optimal β value for a short emission scan duration, a retrospective study was conducted on 57 patients with pathologically confirmed malignancy who underwent whole-body FDG PET/MRI. After applying exclusion criteria, 49 patients (16 men and 33 women; age range, 36–82 years; mean age, 66.1 ± 11.4 years) were included in the patient-based study. A total of 253 lesions were identified in these patients, with a maximum of five lesions randomly selected per patient to avoid statistical clustering bias during evaluation. Out of the 253 lesions, 173 were selected for analysis.

To determine the optimal β values, this retrospective study evaluated three different emission scan durations (2.5 min, 1.5 min, and 1.0 min) and four candidate β values determined by the phantom study. The PET list mode reconstruction was performed with point-spread function recovery, and noise-equivalent counts per axial length (*NEC_patient_*) and noise-equivalent count density (*NEC_density_*) were measured for each scan duration.
(7)$$NEC_{patient}=\frac{\sum_{i=1}^{n}NEC_{i}}{x/100}\;\;\mathrm{[Mcounts/m],}$$
(8)$$NEC_{density}=\frac{\sum_{i=1}^{n}NEC_{i}}{V_{patient}}\times 1000$$
where *P_i_* [Mcounts] is the number of prompt coincidence counts in each bed *i*, *R_i_* [Mcounts] is the number of random coincidences counts in each bed *i*, *n* is the number of beds in the evaluated area (excluding the brain and bladder areas), and *x* [cm] is the imaging length in centimeters. *NEC_i_* was determined by the equation
(9)$$NEC_{i}=\frac{{\left(1-SF\right)}^{2}{\left(P_{i}-R_{i}\right)}^{2}}{\left(P_{i}-R_{i}\right)+\left(1+k\right)R_{i}}\;\;\mathrm{[Mcounts],}$$
where *SF* is the single-scatter fraction, k is a coefficient based on the correction method for random coincidence counts, and variables are in units of million counts.

*NEC_density_* was calculated as
(10)$$NEC_{density}=\frac{\sum_{i=1}^{n}NEC_{i}}{V_{patient}}\times 1000\;\;[{\mathrm{kcounts/cm}}^{3}]$$
where *V_patient_* [cm^3^] is the body volume within the imaging range.

For the evaluation of normal origin image quality, the liver signal-to-noise ratio (*LiverSNR*) was calculated. To evaluate lesion image quality, the maximal standardised uptake value (*LesionSUVmax*), signal-to-background ratio (*LesionSBR*), and *LesionSNR* were measured. The following equations were used to calculate each parameter:(11)$$LiverSNR=\frac{C_{liver}}{SD_{liver}},$$
(12)$$LesionSUVmax=\frac{maximum\;tissue\;activity\;\left[Bq/\;mL\right]}{injected\;dose\;\left[Bq\right]/body\;weight\;\left[g\right]},$$
(13)$$LesionSBR=\frac{LesionSUVmax}{C_{liver}},$$
(14)$$LesionSNR=\frac{LesionSUVmax}{SD_{liver}/C_{liver}}.$$

Here, *C_liver_* and *SD_liver_* represent the average and standard deviation of the SUV in three 1.5 cubic centimeter volumes of interest placed on the normal liver.

Moreover, two readers used a 5-point visual score system to evaluate the image quality of BPL_1.5_ and BPL_1.0_ compared to OSEM_2.5_ based on lesion delineation and the uniformity of physiological uptake in the liver.

The study compared all parameters of BPL_1.5_ and BPL_1.0_ with OSEM_2.5_ using Wilcoxon’s signed-rank test and the Bland-Altman plot. For the visual scores of *LiverSNR*, *LesionSBR*, and *LesionSNR*, a non-inferiority test was performed for each 95% confidence interval (CI) against OSEM_2.5_ using a non-inferiority margin of 10%. Statistical significance was set at *p* < 0.05.

#### 2.3.4. Clinical Evaluation for Detection and Differentiation of Lesions

To validate the optimal β values of BPL and abb-MRI for detection and differentiation capabilities, a total of 163 patients with confirmed malignancy were retrospectively evaluated. These patients were separate from those in the previous study. After applying the exclusion criteria, 157 patients (56 men and 101 women; age range, 32–85 years; mean age, 63.7 ± 11.8 years) were enrolled in the patient-based study. Each examination was assessed separately in four regions (head and neck, chest, abdomen, and pelvis) to avoid statistical clustering bias.

To evaluate the ability of BPL with a determined β value to detect and differentiate lesions, images obtained from OSEM and BPL at different scan durations (OSEM_2.5_, BPL_1.5_ and BPL_1.0_), as well as their fused images with std-MRI and abb-MRI, were visually evaluated by two trained readers using 5-point visual scores. For lesion detection, the 5-point scores were defined as follows: (1) no lesion, (2) possible existence of lesion on PET or MRI, (3) equivocal, (4) possible existence of lesion on PET and MRI, and (5) complete existence of lesion. For lesion differentiation between benign and malignant varieties, the 5-point scores were defined as follows: (1) definitely benign, (2) probably benign, (3) equivocal, (4) probably malignant, and (5) definitely malignant. The reference standards were established using histopathological results and/or follow-up PET/MRI and PET/CT scans during the 6-month follow-up period.

Statistical comparisons were conducted to evaluate the detection and differentiation capabilities of BPL and OSEM. The Wilcoxon signed-rank test was used to compare the two methods, while receiver operating characteristic (ROC) curve analysis and the DeLong test were used to assess differentiation capability. To evaluate inter-reader agreement, Cohen’s kappa coefficients were used. Additionally, a non-inferiority test was performed for each 95% CI against OSEM_2.5_, with the non-inferiority margin set at 2%. For region-based analysis, a *p*-value of 0.0125 was used.

All statistical analyses were performed using MedCalc^®^ Statistical Software version 20.218 (MedCalc Software Ltd., Ostend, Belgium; https://www.medcalc.org; (accessed on 31 March 2023)).

## 3. Results

### 3.1. Phantom Study

The *NEC_phantom_* values were 21.0 [Mcounts] for the 2.5-min scan duration, 12.6 [Mcounts] for the 1.5-min scan duration, and 8.44 [Mcounts] for the 1.0-min scan duration (Table 1). The count for the 1.0-min scan duration was below the clinically recommended value in the Japanese guidelines (>10.8 [Mcounts]).

The results of *BV*, *RC*, *CR*, and visual scores are summarised in Table 2. The *BV* decreased as β values increased. At BPL_1.0_, all β values showed higher *BV* than OSEM_2.5_. At BPL_1.5_, the *BV*s were lower than OSEM_2.5_, with β values of 800 and above. The *CR* was highest at BPL_1.0_, with β values of 500 and 600, followed by 700. On BPL_1.5_, the *CR* was highest with β500, followed by β400 and β600. The *RC* decreased as β values increased but was higher than OSEM_2.5_ at BPL_1.0_ with β values below 700 and at BPL_1.5_ with β values below 600. The visual scores at BPL_1.5_ and BPL_1.0_ with β values of 600–800 were the highest among the β values assessed for all three readers (Figure 2).

According to these findings, β values ranging from 500 to 800 met the criteria. Therefore, for the subsequent clinical evaluation, the four β values of 500, 600, 700, and 800 were selected.

### 3.2. Clinical Evaluation

#### 3.2.1. Clinical Evaluation for Optimal β Value

The primary malignancies of the assessed 49 patients were as follows: gynecological cancers, 12; head and neck cancers, 5; pancreatic cancers, 4; gastrointestinal cancers, 4; malignant lymphoma, 2; prostate cancers, 2; bone and soft-tissue malignancies, 1.

*NEC_patient_* and *NEC_density_* for all assessed emission scans were higher than the recommended guidelines (*NEC_patient_* > 13 [Mcounts/m] and *NEC_density_* > 0.2 [kcounts/cm^3^], respectively)) (Table 3).

The LiverSNR at BPL_1.5_ with β500–800 and BPL_1.0_ with β600–800 were higher than that of the non-inferiority margin (10%) for OSEM_2.5_ (Figure 3).

The Bland-Altman plot showed the smallest mean difference in LesionSUVmax between OSEM_2.5_ and BPL_1.5_ with β600 (Figure 4) and between OSEM_2.5_ and BPL_1.0_ with β700 (Figure 5).

LesionSBR at BPL_1.5_ with β500–800 and BPL_1.0_ with β500–800 were higher than that of the non-inferiority margin (−10%) for OSEM_2.5_ (Figure 6). Similarly, LesionSNR at BPL_1.5_ with β500–800 and BPL_1.0_ with β500–800 was higher than that of the non-inferiority margin (−10%) for OSEM_2.5_ (Figure 7).

The image quality scores of two readers at BPL_1.5_ with β500–800 and BPL_1.0_ with β600–800 were higher than that of the non-inferiority margin (−10%) for OSEM_2.5_. (Figure 8).

From the results above, the optimal β values for BPL_1.5_ and BPL_1.0_ for clinical evaluation were expected to be 600 and 700, respectively; therefore, these values were adopted in the following study (Figure 9).

#### 3.2.2. Clinical Evaluation for Detection and Differentiation of Lesions

The primary malignancies of the assessed 157 patients were as follows: gynecological cancers, 64; head and neck cancers, 42; bone and soft-tissue malignancies, 16; gastrointestinal cancers, 13; pancreatic cancers, 9; malignant lymphoma, 7; bile duct cancers, 3; breast cancers, 2; prostate cancers, 1; malignant melanoma, 1; plasmacytoma, 1; mesothelioma, 1. Three patients had multiple cancers. The area under the curve (AUC) of ROC for the detection capabilities of the combination of OSEM_2.5_/abb-MRI, BPL_1.5_/abb-MRI, and BPL_1.0_/abb-MRI was not significantly different from that of OSEM_2.5_/std-MRI and exceeded the non-inferiority margin (2%) for both readers (Figure 10).

Additionally, there was no significant difference in the AUC for the differentiation capability between BPL_1.5_/abb-MRI, BPL_1.0_/abb-MRI, OSEM_2.5_/abb-MRI, and OSEM_2.5_/std-MRI for both readers (reader 1: *p* = 0.213, *p* = 0.216, and *p* = 0.216; reader 2: *p* = 0.857, *p* = 0.889, and *p* = 0.856, respectively), as illustrated in Figure 11. The 95% CI of the AUC difference between BPL_1.5_/abb-MRI, BPL_1.0_/abb-MRI, OSEM_2.5_/abb-MRI, and OSEM_2.5_/std-MRI for both readers was a fixed margin of <−2% for the non-inferiority test (Figure 12).

Moreover, the inter-reader agreements for both evaluations were substantial, with a weighted kappa of 0.89 and 0.85, respectively.

## 4. Discussion

The study demonstrated that the combination of optimised BPL and abb-MRI in 1.5- or 1.0-min scans per bed position provided image quality and clinical diagnostic performance equivalent to those obtained with the 2.5-min scan using the OSEM and std-MRI protocols. This is the first study to demonstrate the effectiveness of BPL and abb-MRI in performing rapid whole-body FDG PET/MRI through phantom and clinical evaluations.

Shortening the PET emission scan time is one way to overcome the drawbacks of the long examination time of PET/MRI. However, this reduction in scan time results in a lower count of photons, which in turn deteriorates image quality. One method to address this issue is using BPL, an iterative reconstruction method that enables full convergence without an increase in noise by inserting a regularization process in the iterative loop. When the BPL method with an optimal β value is applied, image quality can be improved and quantitative and diagnostic performance can be maintained [13,14,15,16]. Previous studies have also shown that the BPL method can enhance the image quality of low-count images when dynamic acquisition is performed or the administration dose is reduced [17,18,19,20]. Yosii et al. conducted a study using an ^18^F-NaF PET/CT phantom and found that images reconstructed with BPL had better SNR and SBR than those reconstructed with OSEM, even with a shorter scan duration of 90 sec/bed, by optimizing β values [21]. The β value is the only variable parameter in the BPL method; it determines the degree of noise regularization in the iterative loop of image reconstruction but it also affects image contrast and quantitative values. Therefore, appropriate β values must be chosen when using BPL as an alternative to conventional methods [22,23]. In another study, an optimal β value of 600 was suggested for detectability and reproducibility in ^68^Ga-PSMA PET/CT scans, although the study was not intended to shorten scanning duration [24]. Based on the results of both the phantom and clinical studies assessing different β values, the values of 600 and 700 for 1.5-min and 1.0-min emission scans per bed position, respectively, were found to be equivalent in image quality to a 2.5-min emission scan with OSEM reconstruction. This alternative method showed acceptable image quality, quantitative accuracy, and diagnostic performance, making it a viable option for PET/MRI scanning. Additionally, this time reduction can save up to 12 min for whole-body PET/MRI scans compared to the standard 2.5-min emission scan per bed position (or 5 min for respiratory-gated beds) when imaging six beds, two of which are respiratory-gated.

Despite the limitations in MRI sequences and information provided by abb-MRI due to the shortened PET emission scan duration, the diagnostic performance for lesion detection and differentiation between benign and malignant lesions was not inferior to the standard method, even without T2WI. Studies by Kim et al., Yokoo et al., and Yamaguchi et al. have shown that abb-MRI can provide sensitivity comparable to std-MRI in detecting breast cancer, early-stage hepatocellular carcinoma in patients with compensated cirrhosis, and liver metastases in patients with pancreatic ductal adenocarcinoma, respectively, with superior specificity in some cases [7,8,9]. In a systematic review and meta-analysis conducted by Kang et al., abb- and std-MRI showed similar diagnostic efficacy in prostate cancer [10]. Similarly, previous studies have shown that abb-MRI has equivalent diagnostic performance to std-MRI in breast, hepatic, prostate, and other cancers. In this study, whole-body PET/abb-MRI showed comparable detection and differentiation capabilities to PET/std-MRI, and although there were a few discrepancies between the two methods, they were statistically insignificant. Regional analysis revealed discrepant findings between PET/abb-MRI and PET/std-MRI in the abdominal region, possibly because lesions with limited uptake in PET could only be detected by T2WI in std-MRI (Appendix A). Therefore, omitting T2WI may not be the recommended option when evaluating certain areas, such as the abdominal region. Because the inclusion of simultaneously acquired MRI sequences taking longer than the PET emission time in a whole-body PET/MRI protocol would result in a prolonged imaging time, further acceleration of MRI is warranted.

We present certain limitations to our study here. The small sample size (of the clinical evaluations) may have affected the statistical power of our results. The clinical evaluations were performed retrospectively and in a single (internal) institution. We only evaluated malignancy using FDG. Therefore, further evaluations with larger study populations and different PET tracers are required. In addition, our preliminary findings must be validated externally to assess the clinical utility of whole-body PET/abb-MRI in a broader range of applications.

## 5. Conclusions

A combination of TOF-BPL with an optimal β value and abbreviated MRI, which is a shortened version of standard MRI, enabled a rapid whole-body PET/MRI scan in <1.5 min per bed position while maintaining a capability for lesion differentiation equivalent to conventional TOF-OSEM_2.5_/standard MRI. This technique will shorten PET/MRI acquisition time for oncology patients, increasing the throughput of PET/MRI and promoting its clinical application.

## Figures and Tables

**Figure 1 diagnostics-13-01871-f001:**
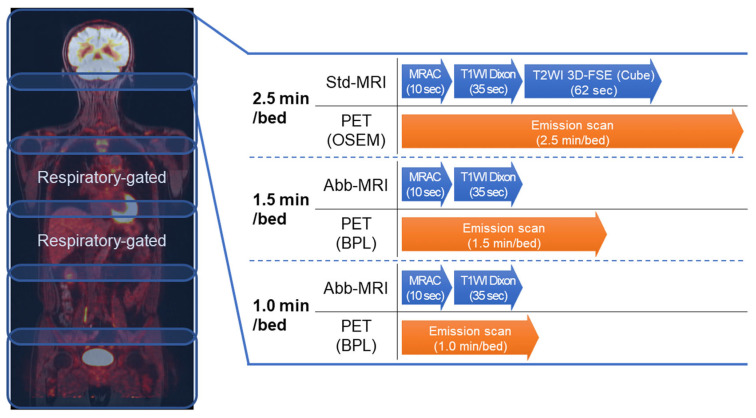
The whole-body positron emission tomography (PET)/magnetic resonance imaging (MRI) scan protocol requires 5 to 6 bed positions per patient to cover imaging from the upper thigh to the top of the head. During the 2.5-min emission scan of PET per one bed position, MRIs including magnetic resonance attenuation correction scan (MRAC), T1-weighted Dixon, and 3D fast spin-echo T2-weighted images (T2WI 3D-Fast Spin Echo (FSE) (Cube)) can be simultaneously acquired. For the 1.5- and 1.0-min scan, only T1-weighted Dixon (T1WI Dixon) can be simultaneously acquired. (BPL, Bayesian Penalised Likelihood).

**Figure 2 diagnostics-13-01871-f002:**
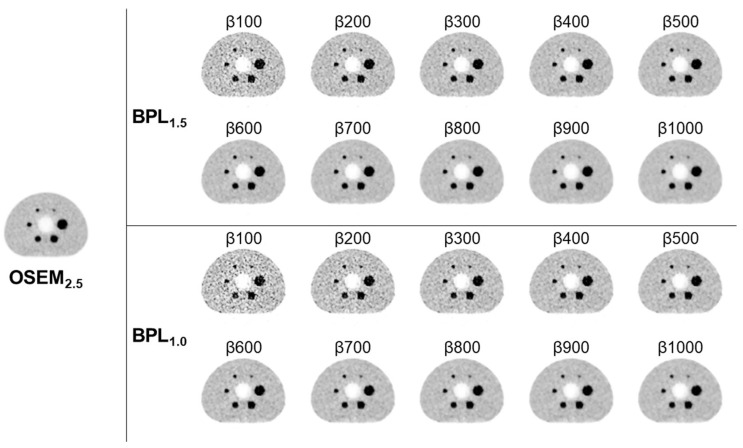
Phantom study images were acquired using the National Electrical Manufacturers Association (NEMA) Image Quality (IQ) Phantom. In Bayesian penalised likelihood reconstruction, lower β values lead to increased background inhomogeneity and decreased delineation of the 10-mm sphere. Conversely, higher β values increase background uniformity, but also result in decreased delineation of the 10 mm sphere. (OSEM, Ordered Subsets Expectation Maximisation; BPL, Bayesian Penalised Likelihood).

**Figure 3 diagnostics-13-01871-f003:**
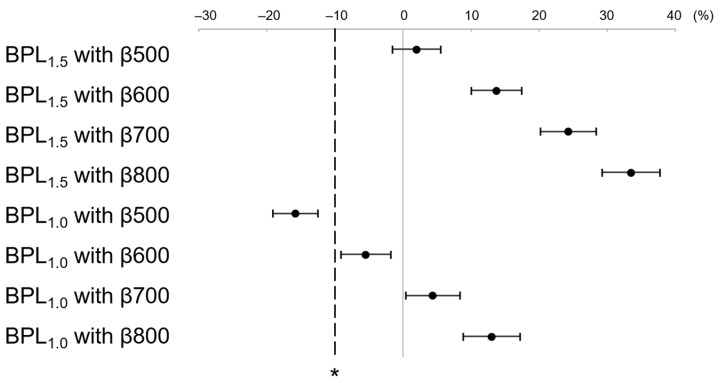
The mean and 95% CI of the percent difference of LiverSNR between OSEM_2.5_ and each assessed reconstruction are shown. The LiverSNR at BPL_1.5_ with β500–800 and that at BPL_1.0_ with β600 and 800 were higher than the non-inferiority margin (indicated by asterisk) for OSEM_2.5_. (OSEM, Ordered Subsets Expectation Maximisation; BPL, Bayesian Penalised Likelihood).

**Figure 4 diagnostics-13-01871-f004:**
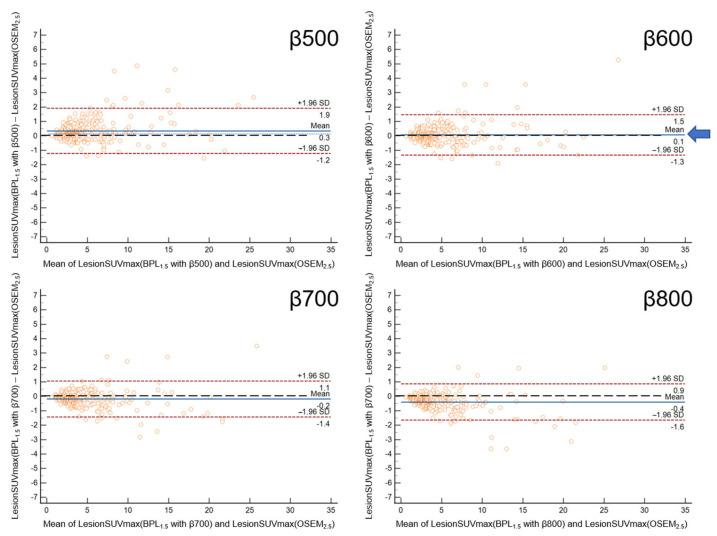
The Bland-Altman plot shows the percent difference in LesionSUVmax between OSEM_2.5_ and the 1.5-min BPL reconstruction. The smallest mean difference in LesionSUVmax between OSEM_2.5_ and BPL_1.5_ was found with a β value of 600 (indicated by the arrow). (OSEM, Ordered Subsets Expectation Maximisation; BPL, Bayesian Penalised Likelihood).

**Figure 5 diagnostics-13-01871-f005:**
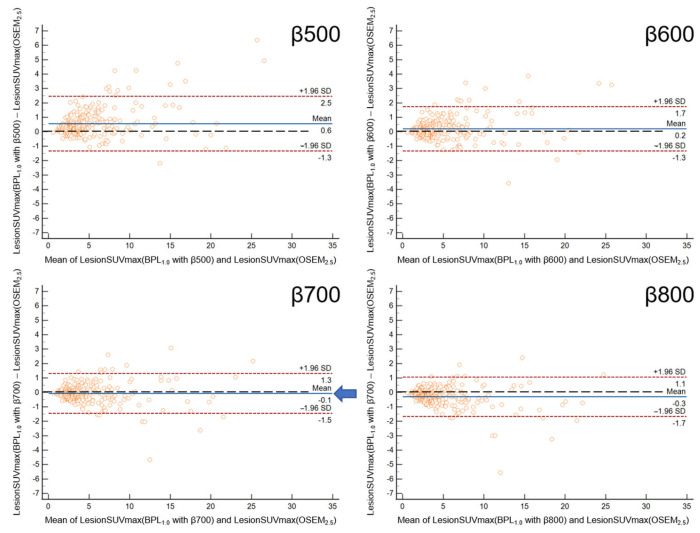
The Bland-Altman plot for the percent difference in LesionSUVmax between OSEM_2.5_ and 1.0 min BPL reconstruction. The smallest mean difference in LesionSUVmax between OSEM_2.5_ and BPL1.0 was found with a β value of 700 (indicated by the arrow). (OSEM, Ordered Subsets Expectation Maximisation; BPL is Bayesian Penalised Likelihood).

**Figure 6 diagnostics-13-01871-f006:**
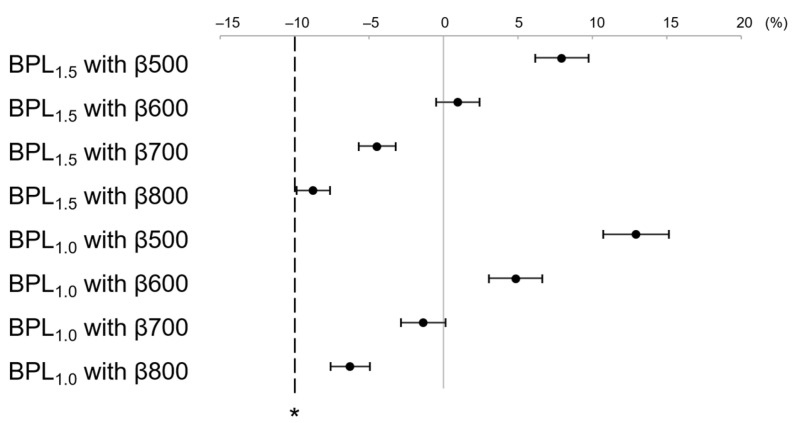
The mean and 95% confidence interval (CI) of the percent difference of LesionSBR between OSEM_2.5_ and each assessed reconstruction. BPL_1.5_ and BPL_1.0_ with β values ranging from 500–800 showed higher LesionSBR than the non-inferiority margin (indicated by asterisk) for OSEM_2.5_. (OSEM, Ordered Subsets Expectation Maximisation; BPL, Bayesian Penalised Likelihood).

**Figure 7 diagnostics-13-01871-f007:**
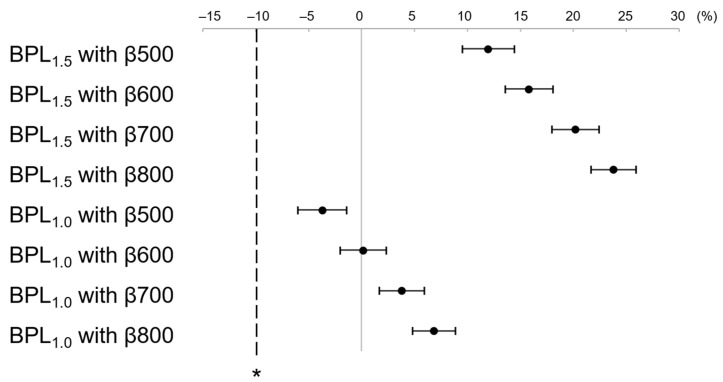
The mean and 95% confidence interval (CI) of the percent difference of LesionSNR between OSEM_2.5_ and each assessed reconstruction. BPL_1.5_ and BPL_1.0_ with β values ranging from 500–800 showed higher LesionSNR than the non-inferiority margin (denoted by asterisk) for OSEM_2.5_. (OSEM, Ordered Subsets Expectation Maximisation; BPL, Bayesian Penalised Likelihood).

**Figure 8 diagnostics-13-01871-f008:**
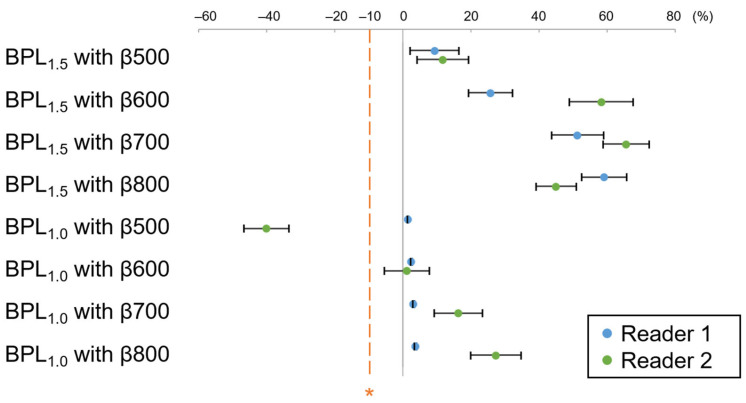
The mean and 95% confidence interval (CI) of the percent difference of the visual scores of each assessed reconstruction. The visual scores for BPL_1.5_ with β values ranging from 500–800 and BPL_1.0_ with β values ranging from 600–800 were higher than the non-inferiority margin (indicated by asterisk) for OSEM_2.5_. (OSEM, Ordered Subsets Expectation Maximisation; BPL, Bayesian Penalised Likelihood).

**Figure 9 diagnostics-13-01871-f009:**
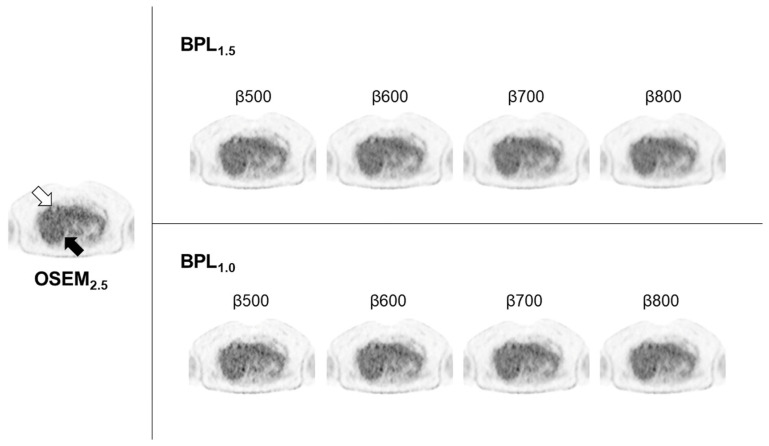
A representative case with liver lesions showing (arrows) the difference in image quality by different scan durations and different β values. The optimal β values based on clinical evaluations were β600 for BPL_1.5_ and β700 for BPL_1.0_. (OSEM, Ordered Subsets Expectation Maximisation; BPL, Bayesian Penalised Likelihood).

**Figure 10 diagnostics-13-01871-f010:**
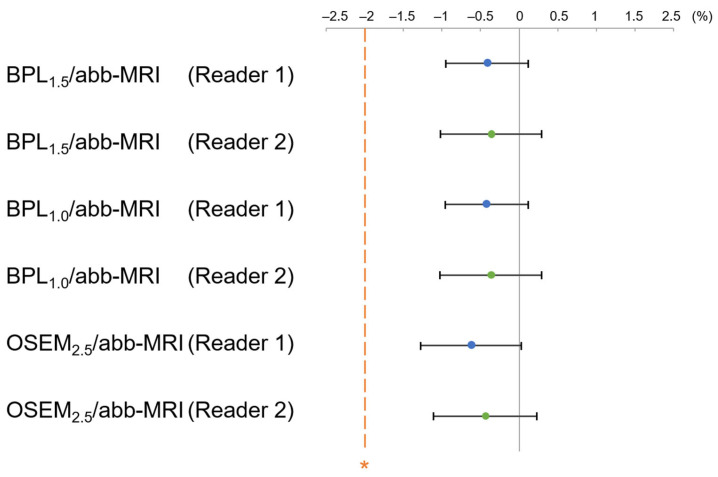
The mean and 95% confidence interval (CI) of the percent area-under-the-curve difference for the detection capability between OSEM_2.5_/std-MRI and each assessed combination of positron emission tomography and magnetic resonance imaging (MRI). The detection capability for OSEM_2.5_/abb-MRI, BPL_1.5_/abb-MRI, and BPL_1.0_/abb-MRI were higher than the non-inferiority margin (indicated by asterisk) for OSEM_2.5_/std-MRI in both readers. (OSEM, Ordered Subsets Expectation Maximisation; BPL, Bayesian Penalised Likelihood).

**Figure 11 diagnostics-13-01871-f011:**
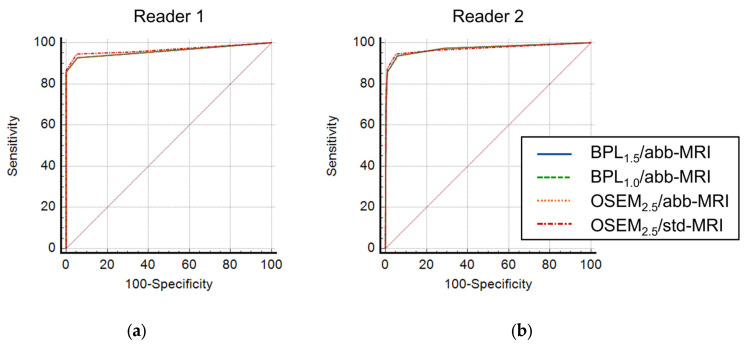
The receiver operating characteristic curve for the differentiation capability between benign and malignant on each combination of PET and MRI. There was no significant difference in the area under the curve for the differentiation capability between BPL_1.5_/abb-MRI, BPL_1.0_/abb-MRI, OSEM_2.5_/abb-MRI, and OSEM_2.5_/std-MRI for reader 1 (**a**) and reader 2 (**b**). (OSEM, Ordered Subsets Expectation Maximisation; BPL, Bayesian Penalised Likelihood.).

**Figure 12 diagnostics-13-01871-f012:**
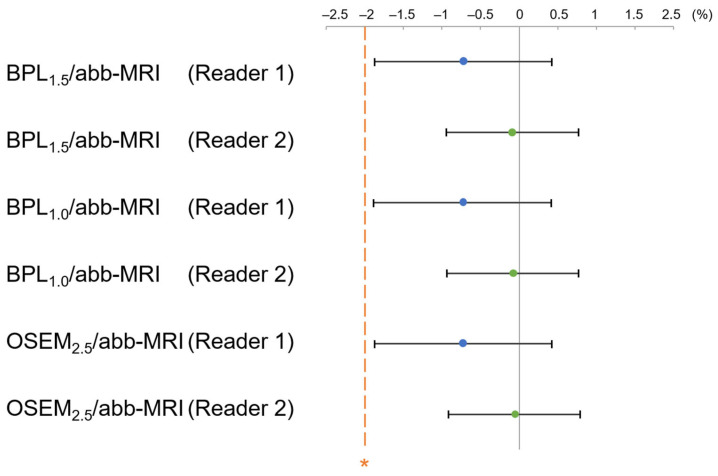
The mean and 95% confidence interval (CI) of the percent area-under-the-curve difference for the differentiation capability between OSEM_2.5_/std-MRI and each assessed combination of positron emission tomography and magnetic resonance imaging (MRI). The differentiation capability of BPL_1.5_/abb-MRI, BPL_1.0_/abb-MRI, and OSEM_2.5_/abb-MRI were higher than the non-inferiority margin (indicated by asterisk) for OSEM_2.5_/std-MRI. (OSEM, Ordered Subsets Expectation Maximisation; BPL, Bayesian Penalised Likelihood.).

**Table 1 diagnostics-13-01871-t001:** The results for the phantom noise-equivalent-count, *NEC_phantom_*_,_ for each scan duration.

Scan Duration	*NEC_phantom_* * [Mcounts]
2.5 min	21.01
1.5 min	12.46
1.0 min	8.44

* The clinically recommended value in Japanese guidelines (>10.8 [Mcounts]).

**Table 2 diagnostics-13-01871-t002:** The results for background variability (*BV*), contrast recovery (*CR*), recovery co-efficient (*RC*), and visual scores for each reconstruction in phantom study. The values in bold are those that satisfy the criteria for determining candidate beta values in each item.

	*BV*	*CR*	*RC*	Visual Scores		
				Reader 1	Reader 2	Reader 3
OSEM_2.5_	7.10	4.44	0.54	2	2	2
BPL_1.5_						
BPL_1.5_ with β100	15.59	3.18	**0.62**	2	2	2
BPL_1.5_ with β200	11.97	3.83	**0.60**	2	2	2
BPL_1.5_ with β300	10.07	4.18	**0.60**	2	2	2
BPL_1.5_ with β400	8.95	**4.33**	**0.58**	2	2	2
BPL_1.5_ with β500	8.20	**4.35**	**0.56**	2	**3**	**3**
BPL_1.5_ with β600	7.65	**4.31**	**0.54**	**3**	**3**	**3**
BPL_1.5_ with β700	7.24	4.23	0.53	**3**	**3**	**3**
BPL_1.5_ with β800	**6.92**	4.14	0.52	**3**	**3**	**3**
BPL_1.5_ with β900	**6.66**	4.03	0.50	**3**	**3**	**3**
BPL_1.5_ with β1000	**6.46**	3.90	0.49	2	2	2
BPL_1.0_						
BPL_1.0_ with β100	18.48	2.48	**0.61**	2	2	2
BPL_1.0_ with β200	13.96	3.11	**0.64**	2	2	2
BPL_1.0_ with β300	11.55	3.50	**0.62**	2	2	2
BPL_1.0_ with β400	10.15	3.69	**0.60**	2	2	2
BPL_1.0_ with β500	9.22	**3.76**	**0.58**	2	2	2
BPL_1.0_ with β600	8.55	**3.76**	**0.56**	**3**	**3**	**3**
BPL_1.0_ with β700	8.05	**3.72**	**0.54**	**3**	**3**	**3**
BPL_1.0_ with β800	7.66	3.66	0.52	2	**3**	2
BPL_1.0_ with β900	7.34	3.58	0.50	2	2	2
BPL_1.0_ with β1000	7.11	3.46	0.49	2	2	2

**Table 3 diagnostics-13-01871-t003:** The results for the patient noise-equivalent-count, *NEC_patient_*, and *NEC_densitiy_* for each scan duration (mean [range], *n* = 49).

Scan Duration	NEC_patient_ ^1^ [Mcounts/m]	NEC_density_ ^2^ [kcounts/cm^3^]
2.5 min	105.64 [63.00–131.65]	1.98 [1.07–2.88]
1.5 min	64.35 [36.34–79.18]	1.23 [0.66–1.91]
1.0 min	43.39 [31.61–65.15]	0.83 [0.44–1.16]

^1^ The clinically recommended value in Japanese guidelines (>13 [Mcounts/m]). ^2^ The clinically recommended value in Japanese guidelines (>0.2 [kcounts/cm^3^]).

## Data Availability

The minimal dataset is within the manuscript. Supporting information files and additional data for analysis are available from the corresponding author. Disclosure of clinical data and images including personally identifiable information is prohibited by the ethical committee in our institution (The Research Ethics Committee of Kobe University Hospital) and by the laws on the protection of personal information in our country. Please direct further data inquiries to the Research Ethics Committee of Kobe University Hospital.

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
