# Peer review of "Rapid Whole-Body FDG PET/MRI in Oncology Patients: Utility of Combining Bayesian Penalised Likelihood PET Reconstruction and Abbreviated MRI"

_diagnostics, 2023, doi:10.3390/diagnostics13111871_

Round 1

Reviewer 1 Report

The article is well structured with fluent language. The methods were described clearly and findings were presented according to guideline. I have only one comments regarding supporting the claim with citation in two parts of the introduction.

1. Please give references for the related studies mentioned for "abbreviated mri"

2.You support your claim with published material, "The hypothesis behind this approach is that “the more MR sequences, the better diagnostic performance” is no longer valid for certain kinds of oncology patients. "

Author Response

We thank the reviewer for the evaluation of our study and the editor for giving us the chance of resubmitting our manuscript. We are happy to address all comments point by point. In addition, the entire text was proofread for English once again. We have made revisions that have been marked in colored fonts for ease of identification in the revised version of our manuscript.

  1. Comments: Please give references for the related studies mentioned for "abbreviated mri"

Response: Thank you for your suggestions. As you pointed out, we have added citations to the references on abbreviated MRI not only in the Discussion but also in the Introduction, and we have added the following article as a reference.

Ref. #8: Takeshi Yokoo, N.M., Neehar D. Parikh, Barton F. Lane, Ziding Feng, Mishal Mendiratta-Lala, Chee Hwee Lee, Gaurav Khatri, Tracey L. Marsh, Kirti Shetty, Colin T. Dunn, Taim Al-Jarrah, Anum Aslam, Matthew S. Davenport, Purva Gopal, Nicole E. Rich, Anna S. Lok, Amit G. Singal. Multicenter Validation of Abbreviated MRI for Detecting Early-Stage Hepatocellular Carcinoma. Radiology 2023, 307, e220917.

  1. Comments: You support your claim with published material, "The hypothesis behind this approach is that “the more MR sequences, the better diagnostic performance” is no longer valid for certain kinds of oncology patients. "

Response: Thank you very much for your valuable comments. As you pointed out, the expression was somewhat subjective, we changed to an objective expression and cited the supporting paper (page 2, line 49-52).

Reviewer 2 Report

Authors in this research presented a Utility of Combining Bayesian Penalized Likelihood PET Reconstruction and Abbreviated MRI.

I have the following major concerns

The state of the art not explored and authors did not justified the reason of this research. 

The authors did not present any model.

Used local dataset to evaluate their case study.

Rejected 

Low quality work ! 

Author Response

We thank the reviewer for the evaluation of our study and the editor for giving us the chance of resubmitting our manuscript. We are happy to address all comments point by point. In addition, the entire text was proofread for English once again. We have made revisions that have been marked in colored fonts for ease of identification in the revised version of our manuscript.

  1. Comments: The state of the art not explored and authors did not justified the reason of this research. 

Response: Thank you very much for your valuable comment. PET image reconstruction using the BPL method remains one of the state-of-art in PET research. It has been actively studied with at least 20 papers published in the past 3 years. Abbreviated MRI is also a recently proposed concept, and while not all relevant papers can be presented here, at least 250 papers have been published in the past three years. The purpose of this study is to shorten the imaging time of whole-body PET/MRI using the BPL method and shortened MRI in order to overcome one of the drawbacks of the integrated PET/MRI system, i.e., the long examination time. We believe that this justifies the study because the shortened imaging time has the advantage of improving throughput, leading to increased use of PET/MRI, as well as improving patient comfort. These aspects are discussed in the introduction section. We also provide below a list of papers on the BPL method that are relevant to this comment.

  1. Tang CYL, et al. Optimization of Bayesian penalized likelihood reconstruction for 68 Ga-prostate-specific membrane antigen-11 PET/computed tomography. Nucl Med Commun. 2023 Jun 1;44(6):480-487.
  2. Naghavi-Behzad M, et al. Comparison of Image Quality and Quantification Parameters between Q.Clear and OSEM Reconstruction Methods on FDG-PET/CT Images in Patients with Metastatic Breast Cancer.J Imaging. 2023 Mar 9;9(3):65.
  3. Young JR, et al. Bayesian penalized likelihood PET reconstruction impact on quantitative metrics in diffuse large B-cell lymphoma. Medicine (Baltimore). 2023 Feb 10;102(6):e32665.
  4. Miwa K, et al. Impact of γ factor in the penalty function of Bayesian penalized likelihood reconstruction (Q.Clear) to achieve high-resolution PET images. EJNMMI Phys. 2023 Jan 22;10(1):4.
  5. Siekkinen R, et al. A retrospective evaluation of Bayesian-penalized likelihood reconstruction for [15O]H2O myocardial perfusion imaging. J Nucl Cardiol. 2023 Jan 19.
  6. Xu L, et al. Phantom and clinical evaluation of the effect of a new Bayesian penalized likelihood reconstruction algorithm (HYPER Iterative) on 68Ga-DOTA-NOC PET/CT image quality. EJNMMI Res. 2022 Dec 12;12(1):73.
  7. Maronnier Q, et al. Evaluation of a method based on synthetic data inserted into raw data prior to reconstruction for the assessment of PET scanners. EJNMMI Phys. 2022 Oct 1;9(1):68.
  8. Tanaka T, et al. Short-time-window Patlak imaging using a population-based arterial input function and optimized Bayesian penalized likelihood reconstruction: a feasibility study. EJNMMI Res. 2022 Sep 8;12(1):57.
  9. Bailly P, et al. Phantom study of an in-house amplitude-gating respiratory method with silicon photomultiplier technology positron emission tomography/computed tomography. Comput Methods Programs Biomed. 2022 Jun;221:106907.
  10. Lohaus N, et al. Impact of Bayesian penalized likelihood reconstruction on quantitative and qualitative aspects for pulmonary nodule detection in digital 2-[18F]FDG-PET/CT. Sci Rep. 2022 May 18;12(1):8308.
  11. Wagatsuma K, et al. Determination of optimal regularization factor in Bayesian penalized likelihood reconstruction of brain PET images using [18 F]FDG and [11 C]PiB. Med Phys. 2022 May;49(5):2995-3005.
  12. Xu L, et al. Small lesion depiction and quantification accuracy of oncological 18F-FDG PET/CT with small voxel and Bayesian penalized likelihood reconstruction. EJNMMI Phys. 2022 Mar 26;9(1):23.
  13. Ribeiro D, et al. Assessing the impact of different penalty factors of the Bayesian reconstruction algorithm Q.Clear on in vivo low count kinetic analysis of [11C]PHNO brain PET-MR studies. EJNMMI Res. 2022 Feb 20;12(1):11.
  14. Dziuk M, et al. Optimal activity of [18F]FDG for Hodgkin lymphoma imaging performed on PET/CT camera with BGO crystals. Nucl Med Rev Cent East Eur. 2022;25(1):47-53.
  15. De Ponti E, et al. Clinical Application of a High Sensitivity BGO PET/CT Scanner: Effects of Acquisition Protocols and Reconstruction Parameters on Lesions Quantification. Curr Radiopharm. 2022;15(3):218-227.
  16. Liu Y, et al. Changes of [18F]FDG-PET/CT quantitative parameters in tumor lesions by the Bayesian penalized-likelihood PET reconstruction algorithm and its influencing factors. BMC Med Imaging. 2021 Sep 16;21(1):133.
  17. Tatsumi M, et al. Effects of New Bayesian Penalized Likelihood Reconstruction Algorithm on Visualization and Quantification of Upper Abdominal Malignant Tumors in Clinical FDG PET/CT Examinations. Front Oncol. 2021 Aug 16;11:707023.
  18. Krokos G, et al. Standardisation of conventional and advanced iterative reconstruction methods for Gallium-68 multi-centre PET-CT trials. EJNMMI Phys. 2021 Jul 17;8(1):52.
  19. Roef MJ, et al. Evaluation of Quantitative Ga-68 PSMA PET/CT Repeatability of Recurrent Prostate Cancer Lesions Using Both OSEM and Bayesian Penalized Likelihood Reconstruction Algorithms. Diagnostics (Basel). 2021 Jun 16;11(6):1100.
  20. Wu Z, et al. Phantom and clinical assessment of small pulmonary nodules using Q.Clear reconstruction on a silicon-photomultiplier-based time-of-flight PET/CT system. Sci Rep. 2021 May 14;11(1):10328. doi: 10.1038/s41598-021-89725-z.
  21. Rijnsdorp S, et al. Impact of the Noise Penalty Factor on Quantification in Bayesian Penalized Likelihood (Q.Clear) Reconstructions of 68Ga-PSMA PET/CT Scans. Diagnostics (Basel). 2021 May 8;11(5):847.
  22. Ribeiro D, et al. Performance evaluation of the Q.Clear reconstruction framework versus conventional reconstruction algorithms for quantitative brain PET-MR studies. EJNMMI Phys. 2021 May 7;8(1):41.
  23. Usmani S, et al. The clinical effectiveness of reconstructing 18F-sodium fluoride PET/CT bone using Bayesian penalized likelihood algorithm for evaluation of metastatic bone disease in obese patients. Br J Radiol. 2021 Apr 1;94(1120):20210043.

  1. Comments: The authors did not present any model.

Response: I regret to say that I did not quite understand what you meant by "model". If you mean a method that serves as a base line to prove a hypothesis, we use the conventional method of OSEM with 2.5-minute acquisition and standard MRI (Page 3, line 118-123; page 5, line 200-204). Also, we use histopathological results and/or follow-up PET/MRI and PET/CT scans during the 6-month follow-up period as a reference standard to evaluate the detection and differentiation between benign and malignancy (Page 6, 221-223).

  1. Comments: Used local dataset to evaluate their case study. 

Response: The issue you comment on is important and is carefully addressed in the discussion section. As you mentioned, this study was conducted at a single institution and no validation was done at an external institution. There are only a few types of PET/MRI systems used clinically, and we hope that the results of our present study are ubiquitous (J Nucl Med. 2015 Feb;56(2):165-8.). However, external validation studies using PET/MRI in different patient backgrounds and at different facilities are considered necessary and are a future challenge.

Reviewer 3 Report

This paper presented a study on the evaluation of diagnostic value of a rapid whole-body fluorodeoxyglucose  PET/MRI approach combining Bayesian penalized likelihood (BPL) positron emission tomography (PET) with optimized β value and abbreviated magnetic resonance imaging (abb-MRI). The study compared the diagnostic performance of this approach with the standard PET/MRI that utilizes ordered subsets expectation maximization (OSEM) PET and standard MRI (std-MRI). From the results, it shows that by combining BPL with optimal β and abb-MRI, rapid whole-body PET/MRI could be achieved in ≤1.5 min per bed position, while maintaining comparable diagnostic performance to standard PET/MRI.    Overall, this paper is well-presented  and could show some detailed analysis. However, the conclusion should be improved. Please explain more the main contribution of this research in the Conclusion part.

Please send this paper for proofreading before it can be published. 

Author Response

We thank the reviewer for the evaluation of our study and the editor for giving us the chance of resubmitting our manuscript. We are happy to address all comments point by point. In addition, the entire text was proofread for English once again. We have made revisions that have been marked in colored fonts for ease of identification in the revised version of our manuscript.

  1. Comments: Please explain more the main contribution of this research in the Conclusion part.

Response: Thank you very much for your comment. The main contribution of this study will be a reduction in PET/MRI imaging time for patients with malignant tumors, which will improve the throughput of PET/MRI and increase its clinical availability. According to your kind suggestion, we added descriptions regarding this issue (Page 15, line 399-404).

Reviewer 4 Report

In this study, authors developed a novel rapid whole-body PET/MRI diagnostic approach based on Bayesian penalized likelihood (BPL) algorithm with optimized β value and abbreviated MRI (abb-MRI). In comparison with traditional OSEM reconstruction PET and standard MRI (std-MRI), BPL PET with abb-MRI exhibited rapid whole-body scanning procedure and non-inferiority imaging diagnostic performance, indicating potential clinical utility in oncology diagnosis. The parameters calculation (eg. BV, CR, RC, NEC, etc.) of optimized β value were solid and well evaluated by phantom study and clinical study. In general, this study is well-designed and good writing. However, some points need to be revised to improve the quality of this manuscript.

1.       In 2.3.1 section, authors mentioned the exclusion criteria but no information of inclusion criteria. Please provide the inclusion criteria of this clinical study.

2.       The basic demographic / clinicopathologic characteristics were lacked in the Result part. It’s recommended to summarize basic information of this clinical cohort at 3.2 section, which is better to show the baseline data of this study.

3.       In Fig 8, there are green and blue dots to show the visual scores distribution. Please add legends of these dots in this figure.

The written language and expression of this manuscript are good. Native speaker revision is recommended.

Author Response

We thank the reviewer for the evaluation of our study and the editor for giving us the chance of resubmitting our manuscript. We are happy to address all comments point by point. In addition, the entire text was proofread for English once again. We have made revisions that have been marked in colored fonts for ease of identification in the revised version of our manuscript.

  1. Comments: In 2.3.1 section, authors mentioned the exclusion criteria but no information of inclusion criteria. Please provide the inclusion criteria of this clinical study.

Response: Thank you very much for your valuable suggestion. We inserted the sentence “Inclusion criteria were patients with current or previous malignancy who had undergone a PET/MRI scan and whose final diagnosis was confirmed by histopathological results and/or follow-up PET/MRI and PET/CT scans during the 6-month follow-up period” in the 2.3.1 section (page 3, line 127-130). We took this opportunity to reconsider the exclusion criteria. Since this study compared images from different image reconstruction conditions within the same patient, there was no need to use blood glucose as an exclusion criterion. Therefore, blood glucose was removed as an exclusion criterion, but as a result, there was no change in the number of eligible cases.

  1. Comments: The basic demographic / clinicopathologic characteristics were lacked in the Result part. It’s recommended to summarize basic information of this clinical cohort at 3.2 section, which is better to show the baseline data of this study.

Response: Thank you very much for your suggestion. As you indicated, we have added the background of each patient in the clinical evaluation by describing the site of each primary tumor and its number (Page 8, line266-267; page12, line 318-319).

  1. Comments: In Fig 8, there are green and blue dots to show the visual scores distribution. Please add legends of these dots in this figure.

Response: Thank you very much for pointing it out. We added the annotations to Fig 8.

Round 2

Reviewer 2 Report

The authors implemented all desired corrections and so accepted.

Acceptable

Reviewer 4 Report

Thanks for the revisions made from the last review comments. Overall, the quality of the current manuscript has improved a lot and meets the criteria for publication of this journal.